# Hydrophobic Nanoporous Silver with ZIF Encapsulation for Nitrogen Reduction Electrocatalysis

**DOI:** 10.3390/molecules28062781

**Published:** 2023-03-20

**Authors:** Yating Qi, Shulin Zhao, Yue Pang, Yijie Yang

**Affiliations:** Tianjin Key Laboratory of Structure and Performance for Functional Molecules, College of Chemistry, Tianjin Normal University, Tianjin 300387, China

**Keywords:** nanoporous silver, ZIF, nitrogen reduction, electrocatalyst

## Abstract

Electrochemical nitrogen reduction reaction (ENRR) offers a sustainable alternative to the environmentally hazardous Haber–Bosch process for producing ammonia. However, it suffers from an unsatisfactory performance due to its limited active sites and competitive hydrogen evolution reaction. Herein, we design a hydrophobic oleylamine-modified zeolitic imidazolate framework-coated nanoporous silver composite structure (NPS@O-ZIF). The composite achieves a high ammonia yield of (41.3 ± 0.9) μg·h^−1^·cm^−2^ and great Faradaic efficiency of (31.7 ± 1.2)%, overcoming the performances of NPS@ZIF and traditional silver nanoparticles@O-ZIF. Our strategy affords more active sites and accessible channels for reactant species due to the porous structure of NPS cores and restrains the evolution of hydrogen by introducing the hydrophobic molecule coated on the ZIF surfaces. Hence, the design of the hydrophobic core–shell composite catalyst provides a valuably practical strategy for ENRR as well as other water-sensitive reactions.

## 1. Introduction

Nowadays, ammonia has become one of the most widely used industrial chemicals for economic development [1,2,3]. However, to date, ammonia synthesis heavily relies on the energy-consuming and capital-intensive Haber–Bosch process [4,5,6,7]. With the reduction in fossil fuels, and the focus on increased greenhouse gas emissions, more sustainable and economical methods of ammonia production are needed to support the growing demand [8]. The electrochemical nitrogen reduction reaction (ENRR) under environmental conditions offers a promising alternative to the highly polluting Haber–Bosch process [9,10,11]. The electrochemical synthesis of ammonia has proposed an electrically driven nitrogen immobilization reaction based on a heterogeneous catalytic process, which not only decreases the high pressure and heat requirements, but also reduces the energy consumption [12,13]. However, it is difficult to effectively improve the ammonia yield for industrial applications with noble metal electrocatalysts [14,15] due to their unsatisfactory catalytic efficiency, cumbersome preparation process, and high cost. Therefore, efficient and stable electrocatalysts are urgently needed to improve the yield and Faradaic efficiency of electrochemical ammonia synthesis. Silver nanoparticles are a kind of common precious metal [16]. Compared with noble metal catalysts such as gold [17,18,19] and ruthenium [20,21,22], silver has an excellent cost advantage. Additionally, silver nanostructures have been proven to efficiently catalyze ENRR [23]. Nevertheless, compared with traditional solid metal nanoparticles, nanoporous structures exhibit numerous interconnected ligaments, which provide more catalytically active sites for boosting the electrocatalysis performance [24,25,26]. Nanoporous gold has been reported for enhancing the ENRR performance with high ammonia yield and efficiency [27,28].

In addition, the unsatisfactory ENRR property also results from the competitive hydrogen evolution reaction (HER) [29,30,31]. Various methods have been utilized to inhibit the occurrence of HER, involving the use of alloy composition [32], fabricating hydrophobic surfaces [33], and coating protective shells [34]. The zeolitic imidazolate framework (ZIF) is one broad class of metal–organic frameworks which is able to immobilize active metal electrocatalysts [35]. Recently, core–shell structures with ZIF encapsulation promote the preferable performance in various areas, involving plastic separation, catalysis, energy-related battery, and gas adsorption [36,37,38]. The stable ZIF structures can prevent the aggregation of active nanostructures, thus sustaining their structural integrity, especially in the application of electrocatalysis. Additionally, the porous structural feature of ZIF can facilitate the concentration of reactant molecules near the catalytic sites, hence promoting the enhancement of catalysis performance for various reactions. Moreover, the hydrophobic post-functionalization of ZIFs can effectively regulate proton availability at the electrode and hence prohibit competing HER [39].

In this work, we propose the design and fabrication of a hydrophobic core–shell nanoporous silver@ZIF-8 nanocomposite with oleylamine surface modification (NPS@O-ZIF, Figure 1). The synergistic interaction between nanoporous silver (NPS) and ZIF shell can reduce the agglomeration of NPS, promote the concentration ability of reactants, and improve the stability of the catalyst. The hydrophobic surface modification by oleylamine can repel protons from the catalyst surface, thus preventing the occurrence of HER. The NPS@O-ZIF composite achieves a high ammonia production rate and Faradaic efficiency ((41.3 ± 0.9) μg·h^−1^·cm^−2^, (31.7 ± 1.2)%) under −1.0 V versus the reversible hydrogen electrode (vs. RHE). The comparison with NPS@ZIF without oleylamine modification indicates that the hydrophobic post-functionalization facilitates ENRR. Additionally, the contrast with Ag@O-ZIF (solid silver nanoparticles@ZIF with oleylamine surface modification) reveals that the porous structure of NPS can boost the catalytic activity by promoting the reaction at the outer and inner surface, bearing numerous active sites. In addition, the NPS@O-ZIF composite also shows excellent stability and selectivity (93%) in the experiments. The above results indicate that the NPS@O-ZIF composite can effectively suppress the HER competition and enhance catalytic activity in the ENRR, providing an effective strategy for the introduction of a core–shell structure and hydrophobic functional layer for water-sensitive electrocatalytic reactions.

## 2. Results

### 2.1. Morphological and Structural Description

A simple method was utilized to fabricate the NPS@O-ZIF composites. Primarily, the NPS cores were synthesized via a dealloying protocol. AgCl templates with a homogeneous cubic structure were synthesized via a reported method wherein the reactant of silver nitrate is ion-exchanged by concentrated hydrochloric acid to form AgCl composition and polyvinylpyrrolidone (PVP) served as a capping agent for stabilizing the formation of AgCl particles [40]. Then, AgCl nanocubes were partially dealloyed by the reducing agent NaBH_4_ and etched by concentrated ammonia with the mass fraction of 25–28% for 1 min to form NPS. The monodisperse NPS nanostructures were characterized by transmission electron microscopy (TEM), exhibiting the size of (232 ± 12) nm and a spherical-like shape but roughened surfaces with interconnected nano-sized channels, which could furnish more plentiful active sites for the later ENRR catalysis (Appendix A). Subsequently, the encapsulation of the ZIF-8 shell was conducted by the crystallization of ZIF in the solvent of methanol in the presence of PVP-capped NPS structures under mild conditions. Additionally, the NPS@ZIF electrode was further functionalized with hydrophobic oleylamine by simply immersing the electrode into oleylamine solution for an optimized duration. Characterization by TEM exhibits the formation of core–shell crystals with the ZIF-8 shell with a thickness of approximately 100 nm (Appendix A). As observed in the high-angle annular dark-field scanning transmission electron microscope (HAADF-STEM) image, a close inspection displays numerous interconnected ligaments with observable voids (Appendix A). It is remarkable that, compared with traditional solid metal nanoparticles, our NPS structures are highly porous with anisotropic morphology, hence greatly increasing the specific surface area. Scanning the transmission electron microscope-energy dispersion X-ray spectroscopy (STEM-EDS) further demonstrates the distribution of each element. The EDS signals of Ag and Cl are acquired from the peaks at 2.99 and 2.65 keV assigned to Ag L and Cl K, respectively. The EDS mappings of NPS clearly state that NPS cores are mainly composed of Ag with a trace residual Cl element with the atomic ratio of Ag:Cl to be 99.7:0.3. NPS@ZIF composites were also characterized for morphology and composition via HAADF-STEM and STEM-EDS. As shown in Figure 1a, the HAADF-STEM image represents the entire and homogeneous encapsulation of ZIF-8 shell on NPS core. The EDS elemental analyses of NPS@ZIF reveal that NPS@ZIF is predominantly composed of silver and zinc elements (Figure 1b,c), which are uniformly distributed throughout the whole surface of the NPS core and NPS@ZIF composite, respectively. EDS mapping signals of Ag L and Zn L are obtained from the peaks at 2.99 and 1.02 keV, from which the atomic ratio of Ag:Zn could be determine to be 41:59, also stating the entire encapsulation of ZIF-8 on NPS cores (Figure 1d).

Furthermore, X-ray photoelectron spectroscopy (XPS) was also performed to analyze the composition and oxidation states in accordance with each element (Appendix A). The XPS survey spectrum of NPS in the Ag *3d* region exhibits two peaks at the binding energies of 368.3 and 374.3 eV corresponding to metallic Ag. The Cl *2p* peak is resolved into two separated peaks at 197.6 and 199.2 eV, confirming the trace quantity of Cl^−^. Based on the integral areas of Ag and Cl peaks, the atomic ratio for Ag:Cl is 98:2 and verifies that the NPS structure is mostly composed of Ag with a trace amount of Cl. NPS@ZIF is further investigated by high-resolution Zn *2p* and Ag *3d* scans (Figure 2a,b). The peaks for the Zn *2p* region exhibit a doublet at 1021.2 and 1044.1 eV, while the two peaks of Ag *3d* are at 197.6 and 199.2 eV. By calculating the integral areas of the corresponding peaks, the Ag/Zn atomic ratio is 5:95. To determine the crystallinity and structure of NPS@ZIF, powder X-ray diffraction (XRD) tests were employed (Figure 2c). The XRD spectra of NPS and NPS@ZIF show five typical diffraction peaks at 38.4°, 44.6°, and 64.7° corresponding to the (111), (200), and (220) lattice planes of the crystalline Ag, which reveal the formation of porous Ag in the NPS@ZIF structures. The XRD pattern of NPS@ZIF also exhibits sharp diffraction peaks corresponding to the highly crystalline ZIF, representing the formation of the ZIF shell in the NPS@ZIF composites. The infrared (IR) spectroscopy of NPS, ZIF, and NPS@ZIF is performed to further represent the formation of the NPS@ZIF structure (Appendix A). The IR spectrum of pristine ZIF exhibits several characteristic peaks. The band at 1584 cm^−1^ is ascribed to the C = N stretch. The intense bands in the region of 1350–1500 cm^−1^ resulted from the entire imidazole ring stretching, while other bands at 900–1350 cm^−1^ and below 800 cm^−1^ are attributed to the in-plane and out-of-plane bending of the ring, respectively. It is worth noting that the band at the same position can also be observed in NPS@ZIF material, proving the formation of the ZIF shell. Additionally, although the characteristic peaks of ZIF overlap with those of NPS cores to some extent, NPS@ZIF also shows consistency in the bands with NPS structures, thereby confirming the NPS cores in the NPS@ZIF composite. Moreover, inductively coupled plasma optical emission spectrometry (ICP-OES) was also used to state the variation of Ag mass composition in NPS and NPS@ZIF materials. After coating the ZIF shell, an Ag mass ratio decreased from 90% to 38%, indicating the successful encapsulation of ZIF on NPS. The porosity of NPS@ZIF was evaluated by a nitrogen gas sorption. Nitrogen adsorption–desorption isotherms of ZIF and NPS@ZIF show a type I curve, stating the formation of ZIF-coated composition. The Brunauer–Emmett–Teller (BET) surface area of NPS@ZIF is 849.2 m^2^/g, while the NPS structure and pristine ZIF material exhibit a BET surface area of 3.8 m^2^/g and 1873.3 m^2^/g. The NPS@ZIF structures show a much higher surface area than NPS particles due to the encapsulation of the porous ZIF shell. Compared with original ZIF crystals, the composite shows a reduction in BET surface areas due to the embedded NPS particles (Appendix A), which are too large to occupy the channels or voids of ZIF structures. The above characterization results quantitatively state the successful formation of the NPS core and homogeneous ZIF shell. Static liquid contact angle tests were conducted to investigate the water affinity of various structures. As shown in Figure 2d, by encapsulating the ZIF shell, the water contact angle increases from (18.7 ± 0.5)° to (61.0 ± 4.2)°. The surface modification of oleylamine further reinforces the hydrophobicity of composites with a high contact angle of (135.0 ± 0.2)°, which proves the successful functionalization of oleylamine.

### 2.2. ENRR Catalysis Performance

We performed the ENRR catalysis tests using the three-electrode electrochemical setup under ambient conditions based on neutral Na_2_SO_4_ solution as electrolyte. Additionally, we used the as-synthesized composites as working electrode with Ag/AgCl and Pt plate as the reference and counter electrode, respectively. All the applied potential was converted into a reversible hydrogen electrode (RHE) scale. The ammonia products were tested via a spectrophotometric indophenol blue approach [41] (Appendix A). The UV–Vis spectra of the electrolyte after 2 h reaction exhibit the absorption band at 655 nm, ascribed to indophenol blue. To exclude the possible contamination, the electrolyte solution is purged with high-purity nitrogen before each measurement. Linear sweep voltammogram (LSV) tests were first conducted ranging from −0.5 V to −1.2 V vs. RHE in the Ar and N_2_ atmosphere. As shown in Figure 3a, the LSV curves of NPS@O-ZIF in Ar- and N_2_- saturated electrolyte display an obvious difference in current density, indicating the remarkable ENRR activity in the potential range from −1.2 V to −0.8 V vs. RHE. Hence, 2 h chronoamperometric (CA) measurements were conducted on NPS@O-ZIF electrodes under the N_2_-saturated electrolyte for 2 h under the applied potentials from −1.2 V to −0.8 V vs. RHE (Figure 3b). The UV–Vis absorption spectra of the electrolyte after the reaction with an indophenol indicator for 2 h exhibit the highest absorbance obtained at −1.0 V vs. RHE, indicating the maximum ammonia yield (Figure 3c). The ammonia yield rate and Faradaic efficiency at different potentials are calculated based on the corresponding absorbance intensity and current density. The highest achieved ammonia yield rate was −1.0 V vs. RHE to be (41.3 ± 0.9) μg·h^−1^·cm^−2^, while the Faradaic efficiency reached (31.7 ± 1.2)% (Figure 3d). One of the by-products of the nitrogen reduction process is hydrazine N_2_H_4_, which could be spectrophotometrically detected utilizing the approach of Watt and Chrisp [42]. Hydrazine is consequently measured under −1.0 V vs. RHE to be (3.0 ± 2.1) μg·h^−1^·cm^−2^. Based on the hydrazine yield, the ENRR selectivity shows excellent value of 93% (Appendix A).

Stability of the electrocatalyst for a long-time reaction is also significant for ENRR application. After 30 h of electrolysis under the potential of −1.0 V vs. RHE, the CA curve of NPS@O-ZIF shows no significant degradation of the current density (Figure 4a), indicating the stable ENRR performance of NPS@O-ZIF composites. Moreover, recycling tests in the N_2_ saturated electrolyte at the potential of −1.0 V vs. RHE was also conducted. Neither the ammonia production rate or the Faradaic efficiency significantly decreased after five-cycle experiments (Figure 4b), stating the excellent durability of NPS@O-ZIF to ENRR. After several long-time ENRR catalytic cycles, NPS@O-ZIF catalysts still maintain the original morphology with no collapsed structures (Appendix A). To further evaluate the superior ENRR property of NPS@O-ZIF composites, a series of control experiments was conducted. Firstly, the ENRR catalytic performance of pristine ZIF structures was likewise evaluated for the CA test under the potential of −1.0 V vs. RHE. ZIF presents the ammonia production rate of (1.7 ± 1.5) μg·h^−1^·cm^−2^ and the Faradaic efficiency of (1.7 ± 1.4)%. A week ENRR performance states the inappreciable activity of a ZIF structure, proving that the active sites of composites for ENRR are mainly derived from NPS cores. Solid silver nanoparticles (Ag) and the corresponding composites (Ag@ZIF, Ag@O-ZIF) were prepared. Since ZIF has a negligible catalytical activity and the source of active sites mainly derives from the NPS cores, the role of NPS was demonstrated by comparing the ENRR performance of NPS@O-ZIF and Ag@O-ZIF, which was tested at −1.0 V vs. RHE under the same condition (Figure 4c). It is evident that the ammonia production rate of NPS@O-ZIF at the potential of −1.0 V vs. RHE ((41.3 ± 0.9) μg·h^−1^·cm^−2^) and Faradaic efficiency ((31.7 ± 1.2)%) is significantly higher than those of Ag@O-ZIF ((17.8 ± 1.7) μg·h^−1^·cm^−2^, (13.7 ± 1.1)%). The ammonia yield rate of NPS@O-ZIF shows a 1.3-fold enhancement and the Faradaic efficiency exhibits 1.3-fold increasement compared with Ag@O-ZIF. This result could be attributed to the porous structure and thus higher density of electrocatalytic active sites. Secondly, NPS@O-ZIF and Ag@O-ZIF show a significantly better ENRR than NPS@ZIF ((24.2 ± 2.5) μg·h^−1^·cm^−2^, (19.7 ± 2.9)%), and Ag@ZIF ((14.1 ± 0.9) μg·h^−1^·cm^−2^, (9.3 ± 2.1)%) under the same conditions. Compared with NPS@ZIF, hydrophobic NPS@O-ZIF reveals a 0.7- and 0.6- fold improvement in accordance with the ammonia yield and catalytic efficiency. In addition, in contrast with Ag@ZIF, the ammonia production rate and Faradaic efficiency of Ag@O-ZIF exhibits 1.3- and 1.5-fold performance. This result proves that the occurrence of HER is suppressed by involving the hydrophobic oleylamine layer, which increases the energy barrier of proton desorption [43]. Moreover, the ENRR performance of NPS was also tested for comparison. The ammonia yield is (7.8 ± 1.4) μg·h^−1^·cm^−2^ and the Faradaic efficiency is (13.8 ± 1.8)%, which are 2.1- and 0.4-fold lower than those of NPS@ZIF, and 4.3- and 1.3-fold lower than those of NPS@O-ZIF, respectively. This comparison states that ZIF encapsulation plays a vital role in reinforcing ENRR by promoting the local concentration of molecules near the active surfaces of electrocatalysts. Hence, the great ammonia yield and Faradaic efficiency prove the feasibility of our design on hydrophobic core–shell NPS@O-ZIF electrocatalysts, attributed to its high stability and excellent electrocatalytic performance.

## 3. Materials and Methods

### 3.1. Materials and General Methods

All available chemicals were purchased commercially and used as received without further purification. AgNO_3_ (AR) from the Beijing Chemical Plant was used during the synthesis. Hydrochloric acid (36–38%), acetone (C_3_H_6_O, ≥99.5%), ethanol (C_2_H_6_O, AR), and sodium sulfate (Na_2_SO_4_, AR) were the products of Tianjin Fengchuan Chemical Reagent Co., Ltd. (Tianjin, China). Polyvinylpyrrolidone (PVP, (C_6_H_9_NO)_n_, average molecular weight = 130,000) was from Shanghai Yuanye Bio-Technology Co., Ltd. (Shanghai, China). Glycol ((CH_2_OH)_2_, AR), isopropyl alcohol (C_3_H_8_O, AR), sodium citrate (Na_3_C_6_H_5_O_7_·2H_2_O, AR), salicylate (C_7_H_6_O_3_, AR), HNO₃ (AR), and sodium borohydride (NaBH_4_, AR) were obtained from Tianjin Damao Chemical Trading Co., Ltd. (Tianjin, China). P-dimethylaminobenzaldehyde (C_9_H_11_NO, AR) was from Shanghai Maclin Biochemical Technology Co., Ltd. (Shanghai, China). NH₃·H₂O (mass fraction = 25–28%) was from Shanghai Meryl Chemical Technology Co., Ltd. (Shanghai, China). Anhydrous methanol (CH_3_OH, AR) was derived from Tianjin Chemical Reagent Plant No. 6. Sodium hydroxide (NaOH, AR) was from Tianjin Kermel Chemical Reagent Co., Ltd. (Tianjin, China). Nitropurna (C_5_FeN_6_Na_2_O, 99.98%) and sodium hypochlorite (NaClO, AR), oleylamine (C_20_H_39_NO_2_, 80–90%) were purchased from Shanghai Yi’en Chemical Technology Co., LTD. Ammonium chloride (NH_4_Cl, 99.99%), hydrazine hydrate (N_2_H_4_·H_2_O, 98%), zinc nitrate (Zn(NO_3_)_2_, 99%), and 2-methylimidazole (C_4_H_6_N_2_, 98%) were purchased from Shanghai Aladdin Biochemical Technology Co., Ltd. (Shanghai, China). Nitrogen N_2_ (≥99.999%) and argon Ar (≥99.999%) gas were from Tianjin Huanyu Co., Ltd. (Tianjin, China).

Transmission electron microscopy (TEM) imaging was performed on a Philips Tecnai F20 system at 200 kV to observe the morphology and quantitatively and qualitatively analyze the elements by EDS mapping. The chemical composition and bonding characteristics were analyzed by X-ray photoelectron spectroscopy (XPS) of PHI Quantera under monochromatic Mg X-ray radiation source. In a Bruker GADDS X-ray diffractometer with Cu Kα radiation, the powder X-ray diffraction (XRD) records the structure of the composite materials. The composition analysis of the materials was also performed using Thermo Scientific iCAP 6500 model ICP-OES. Nitrogen adsorption and desorption tests were performed at 77 K and 1 bar using the Micromeritics ASAP 2020 adsorption apparatus. Nitrogen adsorption–desorption isotherms were analyzed by Micromeritics ASAP 2020 built-in software to obtain specific surface areas and pore sizes. A methanol solution of NPS, NPS@ZIF, or NPS@O-ZIF was drop-casted on silica substrate and dried for static contact angle measurement, which was performed on a Theta Lite tensiometer equipped with the Firewire digital camera. The static contact angle was measured with an ultrapure water droplet (3 μL). Each result was averaged by five tests taken on the NPS, NPS@ZIF, or NPS@O-ZIF functionalized silica substrate.

### 3.2. Synthesis of NPS@O-ZIF

A mixture of silver nitrate and polyvinylpyrrolidone (PVP) solution was prepared and kept stirring under room temperature for 10 min. Hydrochloric acid solution (2.75 mL, 6 mol/L) was added dropwise followed by reaction for 20 min. Then, the mixture solution was washed and centrifuged with water and ethanol several times to obtain the AgCl cubic templates. The precipitate AgCl was dispersed in deionized water to form 10 mg/mL concentration. The solution (5 mL) was mixed with 45 mL glycol under stirring and then NaBH_4_ solution (1 mL, 20 mg/mL) was added dropwise as the reducing agent. After reacting under room temperature for 30 min, ammonia was added to the reaction system and then stirred for 1 min to etch the residual AgCl template. The product solution was washed and successively centrifuged with acetone and methanol. The obtained product was nanoporous silver material (NPS).

Zn(NO_3_)_2_ and 2-methylimidazole were dispersed in methanol solution (25 mM, 10 mL) and then mixed with NPS solution (0.7 mg/mL) for 12 h reaction. After centrifugal cleaning with methanol for several times, the product of NPS@ZIF was obtained and then dispersed in ethanol. The NPS@ZIF solution (4 mg/mL) was distributed on glassy carbon electrode and dried under room temperature. NPS@ZIF functionalized electrode was immersed in oleylamine solution (10 mM, 2 mL) for 40 min and then washed with ethanol before being dried at room temperature. The obtained product was the target NPS@O-ZIF electrode.

### 3.3. Synthesis of Ag Nanoparticles and Corresponding Ag@O-ZIF for Control Experiments

Four grams of PVP with one gram of AgNO_3_ were dissolved in 200 mL glycol under stirring to form the precursor solution. Then, the reaction was kept under the temperature of 160° for 1 h. After the reaction, the mixture was cooled down to room temperature and washed with ethanol several times to obtain Ag nanoparticles. The solution of Zn(NO_3_)_2_ and 2-methylimidazole (25 mM, 10 mL) was then added into the dispersion of solid Ag nanoparticles in the methanol solution (0.7 mg/mL). The encapsulation of ZIF shell was conducted at room temperature. After centrifugal cleaning with methanol several times, the obtained product was Ag@ZIF composite material. Four milligrams of Ag@ZIF material was dissolved in 1 mL ethanol and utilized to functionalize glassy carbon electrode. The Ag@ZIF-modified electrode was immersed in the oleylamine solution (2 mL, 10 mM). Finally, a hydrophobic Ag@ZIF electrode with oleylamine surface modification was obtained for the subsequent electrocatalytic NRR test.

### 3.4. ENRR Tests

Electrochemical tests were made using he electrochemical station (CHI760E) with three-electrode system. The electrolyte was 0.1 M aqueous solution of Na_2_SO_4_. NPS@O-ZIF composite material was used as the working electrode with the Pt plate as counter electrode and Ag/AgCl electrode as reference electrode. The electrochemical system was continuously purged with nitrogen gas for at least 1 h before each ENRR test and then throughout the whole reaction. Cyclic voltammetry measurements were carried out with the applied potentials between −1.2 V and −0.5 V vs. RHE using scanning rate of 50 mV/s in N_2_ saturated electrolyte. Potentiostatic measurements were performed across a range of applied potentials, including −0.8 V, −0.9 V, −1.0 V, −1.1 V, and −1.2 V vs. RHE, at constant room temperature for 2 h. The potential used in this work is converted into reversible hydrogen electrode (RHE) by the equation E(vs. RHE) = E(vs. Ag/AgCl) + 0.0592 × pH + 0.1976. The amount of ammonia generated in the ENRR process was characterized by indophenol blue method. Solution A was a mixture of NaOH (1 M), salicylic acid (5 wt%), and sodium citrate (5 wt%). Solution B is 0.2 M sodium hypochlorite. Solution C is C_5_FeN_6_Na_2_O (1 wt%). Two milliliter electrolyte after ENRR was taken and mixed with the above A, B, and C solutions. After reacting for 2 h, the absorption spectrum exhibits a broad peak at 655 nm. The Watt–Chrisp protocol was utilized to measure the amount of hydrazine. Concentrated hydrochloric acid (30 mL) and ethanol (300 mL) were mixed in a beaker, then 6 g p-(dimethylamino) benzaldehyde was added and the mixture was stirred until the solid was completely dissolved. After electrocatalyzing NRR, the electrolyte was mixed with the above mixture. After reacting for 20 min, the absorption spectrum was measured to determine the amount of hydrazine.

### 3.5. Calculation of Ammonia Yield and Faradaic Efficiency

The amount of ammonia produced during ENRR tests was estimated according to the standard curve using the absorption intensity at 655 nm. Additionally, the corresponding ammonia yield rate (v_NH3_) and Faradaic efficiency (FE) were calculated based on the following equation:v_NH3_ = n_NH3_/(t × A_cat._)
FE = 3F × n_NH3_/Q
where F is Faradaic constant, n_NH3_ is the amount of generated ammonia, Q is the total electric quantity of the whole ENRR process, t is the electrochemical reaction time, and A_cat._ is the loading area of catalyst.

## 4. Conclusions

In this work, a core–shell composite structure with hydrophobic surface modification was successfully synthesized for catalyzing ENRR. The morphology, composition, and surface property of the prepared catalysts were well characterized and discussed. Our design of NPS@O-ZIF achieves a high ammonia yield of (41.3 ± 0.9) μg·h^−1^·cm^−2^ and a great Faradaic efficiency of (31.7 ± 1.2)%. The comparison of the ENRR performance with the Ag@O-ZIF and NPS@ZIF demonstrates the significance of active sites on the NPS cores and the suppression of HER by utilizing the oleylamine layer. This work effectively solves the bottleneck problems of ENRR catalysis and will create huge opportunities for sustainable ENRR applications.

## Data Availability

The data presented in this study are available in the Appendix A.

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
