# Peer review of "Hydrophobic Nanoporous Silver with ZIF Encapsulation for Nitrogen Reduction Electrocatalysis"

_molecules, 2023, doi:10.3390/molecules28062781_

Round 1

Reviewer 1 Report

The manuscript by Yang et al. presents a hydrophobic core-shell NPS@O-ZIF composite catalyst for electrochemical nitrogen reduction reaction, in which nanoporous silver is incorporated in a porous ZIF shell with post hydrophobic functionalization. Actually, electrochemical NRR is a promising approach to ammonia production under ambient condition and has recently become a challenging subject. The designing strategy and the excellent performances for the composite catalyst make it a remarkable system for future applications. The NRR performances have also been well illustrated by a series of controlling experiments. 

1. The NRR performance for core-shell NPS@O-ZIF composite is superior, compared with traditional silver nanoparticles@ZIF composite and individual NPS catalyst. A quantitative description is recommended to make it clearer to reader.

2. The NRR performance of NPS@O-ZIF should also be compared with pure ZIF.

3. The details for contact angle measurement of NPS, NPS@ZIF and NPS@O-ZIF are missing.

Reviewer 2 Report

The MS molecules-2285255 represents the core-shell nanoarchitecture built from the nanoporous Ag-based core and nanoporous ZIF-based shell. It is worth noting the simplicity of the proposed technique, which, in turn, is based on self-organization via the coordinative bonds. The hydrophobic coating of the nanostructures also derive from the coordinative binding of the oleates with Zn2+ ions. This strategy differs from the common approaches based on the construction of ZIF-based structures followed by the their loading with the small sized nanomaterial. The excellent catalytic activity of the developed nanomaterial confirms its nanoporous structure. 

Some minor changes are required before the acceptance.

1. line 63, the sentence lacks for "that"

2. No citing on Figures 1a and 1b is provided.

3. the concentration of ammonia and the time duration required for the efficient etching of AgCl nanoparticles should be specified.

4. the role of the addition of the reductant NaBH4 to the synthesis of AgCl nanoparticles must be discussed. It seems as if the Ag-nanoparticles derive from the presence of the reductant. This disagrees with the discussion of the results, where the role of NaBH4 is not discussed.

Reviewer 3 Report

1.          Please provide the PXRD and BET for the composition

2.             The manuscript contains spelling/grammatical errors. So, the language should be polished thoroughly.

3.          Why not do the ENRR Catalysis Performance on ZIF-8 OR NPS?

4.          In the introduction, the authors have to introduce the structural feature, such as Chemosphere, 2022, 307,135729; ACS Appl. Mater. Interfaces., 2021, 13, 12463−12471and Dalton Trans., 2021, 50, 18016–18026.

5.          Please provide the IR analysis

6.          Give the mechanism of ENRR Catalysis Performance

Round 2

Reviewer 3 Report

accepted.